# A Revised View of the *LSU* Gene Family: New Functions in Plant Stress Responses and Phytohormone Signaling

**DOI:** 10.3390/ijms24032819

**Published:** 2023-02-01

**Authors:** Javier Canales, Anita Arenas-M, Joaquín Medina, Elena A. Vidal

**Affiliations:** 1Instituto de Bioquímica y Microbiología, Facultad de Ciencias, Universidad Austral de Chile, Valdivia 5110566, Chile; 2ANID-Millennium Science Initiative Program-Millennium Institute for Integrative Biology (iBio), Santiago 8331150, Chile; 3Centro de Biotecnología y Genómica de Plantas, INIA-CSIC-Universidad Politécnica de Madrid, 28223 Madrid, Spain; 4Centro de Genómica y Bioinformática, Facultad de Ciencias, Universidad Mayor, Santiago 8580745, Chile; 5Escuela de Biotecnología, Facultad de Ciencias, Universidad Mayor, Santiago 8580745, Chile

**Keywords:** sulfate deficiency, LSU, response to low sulfur, abiotic stress, sulfur nutrition, ethylene, auxin, phytohormones, transcription factors

## Abstract

LSUs (RESPONSE TO LOW SULFUR) are plant-specific proteins of unknown function that were initially identified during transcriptomic studies of the sulfur deficiency response in Arabidopsis. Recent functional studies have shown that LSUs are important hubs of protein interaction networks with potential roles in plant stress responses. In particular, LSU proteins have been reported to interact with members of the brassinosteroid, jasmonate signaling, and ethylene biosynthetic pathways, suggesting that LSUs may be involved in response to plant stress through modulation of phytohormones. Furthermore, in silico analysis of the promoter regions of *LSU* genes in Arabidopsis has revealed the presence of cis-regulatory elements that are potentially responsive to phytohormones such as ABA, auxin, and jasmonic acid, suggesting crosstalk between LSU proteins and phytohormones. In this review, we summarize current knowledge about the *LSU* gene family in plants and its potential role in phytohormone responses.

## 1. Introduction

Sulfur (S) is an essential macronutrient required for plant growth because it is a constituent of relevant biomolecules such as the amino acids methionine and cysteine, the antioxidant glutathione, coenzymes and prosthetic groups [1]. Therefore, plants cannot adequately complete their life cycle when subjected to an S-deficiency condition [2]. The symptoms mainly appear in the young parts of the plant and are characterized by reduced height, chlorosis of the leaves, and accumulation of anthocyanins [1,3]. Unlike other nutritional deficiencies, S deficiency typically results in reduced shoot growth compared to root growth [3].

At the molecular level, the response to S-deficiency can be divided into two main stages based on the duration and severity of the deficiency [3]. In the initial stage, plants alter the expression of primary genes involved in S-assimilation and uptake from the soil, and mobilize stored inorganic S from the vacuole [3,4]. However, if S remains a limiting factor, plants intensify organic S fluxes and activate stress defense responses, followed by the downregulation of genes responsible for nitrogen uptake and assimilation [5].

The advent of transcriptomics studies has allowed considerable progress in the identification of S-responsive genes, mainly in the model plant *Arabidopsis thaliana* [6]. An integrative metaanalysis of transcriptomic data from five different S experiments in public databases uncovered a robust set of genes whose expression depends only on the availability of S in Arabidopsis [7]. Interestingly, the biological function of approximately 45% of these robust S-responsive genes is currently unknown. A small gene family, “*RESPONSE TO LOW SULFUR*” (*LSU*), belongs to this group of consistently S-responsive genes, suggesting that these genes could be an essential component of this nutritional response [7].

Several studies have shown that LSUs are important hubs of protein interaction networks with potential function in the plant stress response [8,9,10]. Phytohormones play a critical role in helping plants adapt to adverse environmental conditions, including abiotic and biotic stresses [11]. Interestingly, it has been reported that LSU proteins interact with members of brassinosteroid signaling [12], jasmonate signaling [8], and the ethylene biosynthetic pathway [13], suggesting that *LSUs* could be involved in the response to plant stress by modulating phytohormones. Furthermore, in silico analysis of the promoter regions of *LSU* genes in Arabidopsis showed that they have cis regulatory elements which are potentially responsive to phytohormones such as ABA, auxin, and jasmonic acid [14]. This evidence suggests a possible crosstalk between LSU proteins and phytohormones.

The molecular functions of LSU proteins remain incompletely understood, but in recent years, several studies have shed light on their putative functions and evolution. In this review, we summarize the current knowledge about the *LSU* gene family in plants and their potential role in phytohormone responses.

## 2. General Features and Evolutionary History of the *LSU* Gene Family

### 2.1. The Discovery of the *LSU* Gene Family

*LSU* genes were first described in the context of the S-deficiency response in *Arabidopsis thaliana* by Maruyama–Nakashita [15]. This study identified *LSU1* and *LSU2* as two of 15 S-responsive genes that were significantly upregulated at multiple time points after plants were transferred to S-free medium [15]. Specifically, *LSU1* was significantly induced 4, 8, 12, and 24 h after S-deficiency, whereas *LSU2* was upregulated at 8, 12, and 24 h, indicating that the response of *LSU* genes to this nutritional deficiency is maintained during the first 24 hours in Arabidopsis roots [15]. In the same year, members of this gene family were also identified in tobacco plants in the context of S-deficiency by using the suppression subtractive hybridization approach [16], suggesting that the response of *LSU* genes to this nutritional deficiency may be conserved in plants.

More recently, S-deficiency has been reported to also induces the expression of the *LSU3 and LSU4* genes in Arabidopsis roots and leaves [17]. However, the degree of induction was not the same among members of this family: the induction of *LSU4* by S-deficiency is lower than *LSU1/2/3* in both organs [17]. Furthermore, this study also showed that the mRNA levels of all tomato *LSU* genes and three wheat *LSU* genes increased by S-deficiency [17], supporting the idea that this gene family is associated with S-deficiency and the response to this nutritional deficiency could be conserved in angiosperm plants.

Analysis of the Arabidopsis genome revealed four members of the *LSU* family (*LSU1–4*), distributed in two chromosomes [14]: *LSU1* and *LSU3* are located on chromosome 3, and *LSU2* and *LSU4* are on chromosome 5. These two pairs of *LSU* genes are separated by a small distance of about 2 Kb [14]. In addition, *LSU* genes are characterized by their small size (approximately 300 bp of coding sequence) and the absence of introns [14].

### 2.2. Evolution of *LSU* Gene Family

The evolutionary history of the LSU gene family has recently been analyzed by using genomic information from 134 plant species that include representatives of the major phylogenetic groups of the *Viridiplantae* clade [17]. The first notable finding of this study was that the *LSU* family probably originated from the common ancestor of seed plants [17]. As shown in Figure 1, no homologous *LSU* sequences were found in the genomes of ancient vascular plants such as *Selaginella moellendorffii*, nonvascular plants, or microalgae. This result contrasts with the evolutionary history of other genes involved in the S-deficiency response, such as sulfate transporters, or genes encoding enzymes of S assimilation, such as ATP sulfurylase (APS) or APS reductase (APR), which are present in all *Viridiplantae* from microalgae to angiosperms (Figure 1) [17]. Furthermore, the family of the central transcriptional regulator of plant S-response, *ETHYLENE-INSENSITIVE3-LIKE3* (*EIL3*), is present in all analyzed land plant genomes (Figure 1), indicating that the evolutionary appearance of *LSU* family is recent compared to other genes involved in the S-deficiency response [17]. In addition, several experimental-verified interactors of *LSU* genes in Arabidopsis, such as *APS1*, *GAPC1*, *RAF2*, *FSD2*, and RAP1, are also present in all analyzed *Viridiplantae* genomes [17].

The number of *LSU* genes varies between angiosperm plants, ranging from 1 to 9 members [17]. This variation is mainly due to genome size, as a significant positive correlation has been found between the number of *LSU* genes and the size of the genome [17]. Furthermore, the analysis of the distribution of normalized *LSU* gene numbers in monocotyledon and eudicotyledons revealed no significant differences between these clades [14], suggesting that the *LSU* family does not expand during the evolution of angiosperm plants. Unlike the *LSU* copy number, the evolutionary distance between *LSU* genes of the same species in monocotyledons is more significant than in eudicotyledons, indicating a potential functional divergence of *LSU* genes within monocotyledon species such as wheat [17].

Phylogenetic analysis revealed that *LSU* genes could be divided into three main phylogenetic groups: Group A, which includes most of the monocotyledon species; Group B, including most of the malvid species; Group C, including most of the rosid species [17]. Protein sequence analysis based on 270 LSU sequences showed that the central region of LSU proteins has two highly conserved domains and also revealed the presence of three additional motifs that further support the classification by phylogenetic analyses [17]. The significance of conserved and group-specific motifs in LSU proteins is currently unknown, and further research should be undertaken to reveal the molecular function of these domains [17].

## 3. Functional Analyses of the *LSU*
Family

### 3.1. Subcellular Localization of LSU Proteins

Biochemical fractionation has shown that LSU1 and LSU2 proteins localize in multiple cell compartments, including nuclear, cytosolic, and microsomal fractions [19], while LSU dimers are most probably located in the cytosol [12]. Data from the SUBA4 database [20] support a mainly nuclear and cytoplasmic localization for LSU1, and nuclear, cytoplasmic, chloroplastic, and mitochondrial localization for LSU2 and LSU3 (data for LSU4 is not available) (from Cell eFP viewer, ePlant, [21]). In tobacco, UP9C has a reported nuclear and cytosolic localization, and, generally, a putative nuclear localization signal has been found in this protein [14]. Although no nuclear localization signal has been found in Arabidopsis LSUs, their small size probably allows them to readily cross the nuclear pore [14].

### 3.2. Different Members of the *LSU* Gene Family Showed Tissue-Specific Expression

Analysis of *LSU* tissue expression has been limited to *LSU1* and *LSU2*, showing that these proteins present specific tissue expressions consistent with a specialized role. For example, *LSU1* is diffusely expressed in roots and strongly expressed in guard cells, indicating a role in stomata function, whereas *LSU2* is ubiquitously expressed in leaves and roots [19]. Additionally, we performed a correlation analysis of *LSU* expression data across 69 samples of the Arabidopsis developmental atlas included in the eFP browser [21,22]. *LSU1*, *LSU2*, and *LSU3* showed a high and significant positive correlation (*p*-value < 0.01, Figure 2A), indicating that they have similar expression patterns in the developmental atlas. In contrast, no significant correlation was found between *LSU4* and *LSU1/LSU2/LSU3*, indicating that Arabidopsis *LSU* genes are grouped into two clusters according to developmental and tissue-specific expression (Figure 2A).

In Figure 2B, we compared the expression patterns of *LSU2* (as a representative gene with a higher average expression of the *LSU1/LSU2/LSU3* cluster) and *LSU4* to illustrate the two different groups of Arabidopsis *LSU*s. In the case of *LSU2*, this gene is mainly expressed in leaf petiole, leaf vein, and pod of the senescent silique 1 (Figure 2B). In contrast, the maximum expression of *LSU4* is detected in floral tissues (Figure 2C), supporting the contrasting correlation values between *LSU4* and other LSUs obtained in Figure 2A. We then asked whether the existence of two contrasting groups of *LSU* expression also occurs in other plant species. To this end, we performed the same analysis with wheat *LSU* genes [17] as an example of a monocotyledon plant. We also found two contrasting groups of *LSU* genes in wheat according to their expression patterns throughout development (71 samples; Figure 2A,C), suggesting a possible functional divergence between members of this family in plants.

### 3.3. Functional Analyses of LSUs in Arabidopsis

Insights into the role of individual LSU proteins in Arabidopsis have been obtained by characterization of available T-DNA insertional lines, mainly for *LSU2* and *LSU4*. Currently, no T-DNA lines for *LSU1* are available, and although insertional lines for *LSU3* exist, no reports have been published to date. The involvement of *LSUs* with biotic stress responses was first suggested in analyses of the Arabidopsis protein-protein interactome, showing that *LSUs* represented hubs in immune response-related networks [9]. Analysis of *lsu2* mutants showed that this protein was necessary for normal immune plant response to the bacterium *Pseudomonas syringae* DC3000 (avrRpt2) and the fungi *Hyaloperonospora arabidopsidis*. *LSU2* was identified as a target of pathogen effector proteins and was proposed to act as part of a growth-suppression mechanism mediated by the *P. syringae* 2 (RPS2) NB-LRR protein [9]. Later work using *lsu2* mutants showed that stomatal closure in response to *P. syringae* DC3000 and the human pathogen *Salmonella enterica subsp. enterica serovar Typhimurium* strain 14028s was significantly reduced.

These results implicate *LSU2* as part of an important plant defense mechanism that occurs in guard cells to prevent the entry of bacterial pathogens [24].

In addition to its role in the immune response to pathogens, *LSU2* works as an integrator of light and chloroplast signaling. *LSU2* is induced by light and lincomycin, a chloroplast biogenesis inhibitor [24]. *lsu2* mutants have more than twofold chlorophyll contents compared to wild-type plants when deetiolation is performed in a wide range of light fluences [25]. As such, *LSU2* (together with six other genes) was classified as an enhanced deetiolation (*end*) gene [25]. Consistent with the putative role of *LSU2* in integrating light and plastid signaling, *lsu2* mutants have a decreased expression of the photosynthesis-related genes *Lhcb1.4*, *RbcS1A*, *PsbS*, and *CHS* [25]. *LSU2* has also been shown to act as part of a common response module of genes involved in plastid performance and retrograde signaling [26]. These functions of *LSU2* are consistent with its subcellular localization in the chloroplast.

Regarding the *LSU4* function, *lsu4* mutants show a late flowering phenotype under short-day conditions, whereas flowers formed in the first flowering phase present aberrant developmental phenotypes and do not produce siliques [27]. This is accompanied by a decrease in the expression of critical flowering genes such as *LFY*, *AP1*, *AP3*, *PI*, and *SEP3* transcripts and an increase in the expression of *AP2*, *AG*, and *SEP2* [27]. Consistent with these phenotypes in the *lsu4* mutant, *LSU4* shows an induced expression during flowering and fruit formation [27]. The induction of *LSU4* is also evident during deficiencies in different nutrients (phosphorous, nitrogen, potassium, iron), indicating a possible role of *LSU4* as a coordinator of nutrient demand and flowering [27].

Given that no individual T-DNA lines exist for all *LSU* genes and to uncover phenotypes that can be masked by potential functional redundancy, Arabidopsis knockdown lines have been generated by using artificial microRNAs (amiRNAs) targeting all *LSU* members (>80% reduction in *LSU1*, *LSU2* and *LSU3* and 50% for *LSU4*) [19]. These lines present no obvious phenotypes when grown in standard soil or in vitro conditions [19]. However, closing of abaxial stomata in response to S-deficiency was impaired in the knockdown lines, leading to increased water loss and indicating a role for *LSUs* in this response [19]. This phenotype is consistent with the expression of *LSU1* in guard cells [19] and the reported role of *LSU2* in stomata closure [24].

Furthermore, H_2_O_2_ production in guard cell chloroplasts of knockdown lines was reduced compared to wild-type plants in response to S deficiency and other stresses such as high salt and Cu [19]. Consistent with this observation, the iron-dependent superoxide dismutase 2 (FSD2) was shown to physically interact with LSU1 and LSU2 in vitro and in vivo, and this interaction was shown to increase the enzymatic activity of FSD2; thus, the production of H_2_O_2_ from O_2_^−^ [19]. Interestingly, the LSU1-FSD2 interaction is targeted and interfered by different virulence effectors, revealing a mechanism used by bacteria to abrogate pathogen-associated molecular pattern-triggered immunity [19]. As expected, the amiRNA lines are more susceptible to pathogen attack, and conversely, *LSU1* overexpressor lines present an enhanced disease resistance phenotype under standard conditions, as well as under conditions of abiotic stress [19]. Interestingly, *LSUs* have also been linked to the function of beneficial bacteria such as *Enterobacter* sp. SA187, an endophytic bacterium that protects plants from abiotic stresses. Plant colonization with SA187 can completely suppress the increased ROS levels and alleviate growth suppression in *LSU* knockdown plants subjected to high salt stress [28].

### 3.4. Functional Analyses of LSUs in Other Plants

Similar to Arabidopsis, UP9 proteins, the *LSU* homologs in tobacco, play an important role in S-deficiency responses [14]. Knockdown of UP9 proteins by using a *UP9C* antisense line alters glutathione levels in roots and mature tobacco leaves, especially under S-deficiency conditions [29]. The effect of *UP9* downregulation is organ-dependent, with mature leaves of *UP9* transgenic plants having higher levels of total S and glutathione (GSH) than wild-type plants in S-deficiency, similar to plants grown in S sufficiency [29]. In the case of roots, total S and glutathione are more affected, presenting significantly decreased levels in the *UP9* transgenics in both S conditions [29]. This resulted in knockdown plants presenting shorter roots, whereas shoot growth was unaffected. At the transcript expression level, *UP9* knockdown resulted in altered levels of S-related enzymes and transporters, as well as genes related to ethylene, jasmonic acid, and polyamines [29]. *UP9* knockdown plants also present altered metabolite profiles under S-deficiency, suggesting UP9s are key to adaptation to S-deficiency conditions [29]. By using these knockdown lines, *UP9* was also shown to be required for the increased ethylene production that occurs during S-deficiency [13]. This is partly due to its interaction with the ACC oxidase protein [13]. *UP9* downregulation affected the S-deficiency response of several genes, mainly involved in S metabolic processes, but also transcription regulation, defense response, and hormonal pathways such as ethylene, ABA, and CK [13].

The response of *LSU* genes to S deficiency has been demonstrated in several crops: tomato [17,30], rice [31], and wheat [17] (Table 1). In the case of wheat and tomato plants, the response of *LSUs* was verified by qPCR after two weeks of S deficiency in roots and leaves [17]. Furthermore, it has recently been reported that the three *LSU* genes of rice were upregulated in response to S deficiency in roots and shoots [31]. Interestingly, one of the *LSU* genes, *Os10g0509600*, was strongly induced by S deficiency but significantly downregulated in both the split-root half with S resupply and the split-root half that remained under S deficiency, suggesting a local and systemic response of this *LSU* gene to sulfate resupply [31]. In addition to these species, it has been proposed through mRNA-protein network analysis that *LSU* genes are also involved in the response to S deficiency in pea seeds [32] (Table 1).

*LSU* genes have also been described in the context of response to fungus infection [33] (Table 1). Recent research uncovered the dynamics of the transcriptome in sugarcane infected with *S. scitamineum* and identified an *LSU* homolog gene (*S.off_newGene_71819*) as a highly connected gene from a coexpression module linked to the metabolism associated with the defense response, suggesting an important role for this *LSU* in the plant–pathogen interaction [33]. Interestingly, the expression levels of this *LSU* homolog gene were higher in a sugarcane genotype resistant to *S. scitamineum* infection compared to a susceptible genotype. Together, these results suggest that *LSU* genes might play an important role in the defense response of sugarcane to fungus.

Glucosinolates are sulfur- and nitrogen-containing secondary metabolites of *Brassicaceae* plants that play an important role in plant defense by acting as a deterrent to herbivores and pathogens [35]. Interestingly, a recent genome-wide association study in *Brassica juncea* discovered that *BjuA033112*, a homolog gene of the Arabidopsis *LSU2,* is significantly associated with the gluconapin content, one of the main glucosinolates [34]. These findings suggest that *LSU* genes may be involved in the biosynthesis of glucosinolates in *Brassica juncea*.

## 4. LSU Protein Interactions and Phytohormone Signaling

### 4.1. LSU Protein Interactions Suggest Some Degree of Specialization within this Family in Arabidopsis

Computer modeling of different LSU proteins identified coiled-coil motifs in their structure [14]. Additional circular dichroism studies using a recombinant UP9C protein suggest that this protein is mostly alpha-helical, which further supports a coiled-coil structure [29]. Three-dimensional structure prediction by using AlphaFold is also consistent with an alpha-helical structure for Arabidopsis LSUs (https://alphafold.ebi.ac.uk/ (accessed on 21 November 2022); Q9SCK1, Q9FIR9, Q9SCK2, and Q8L8S2 for the LSU1-4, respectively). Although the 3D structure of LSUs has not been experimentally determined, the presence of a coiled-coil motif, which facilitates oligomerization, indicates that these small proteins can form multimers and interact with multiple kinds of proteins. Consistently, BiFC analyses have shown *in planta* formation for Arabidopsis LSU2, LSU3, and LSU4 homodimers [12]. However, Y2H analyses have confirmed interactions only for LSU1-LSU1 and LSU2-LSU2 homodimers, and for LSU1-LSU2, LSU1-LSU3, and LSU1-LSU4 heterodimers [12]. This suggests that the efficiency of interaction may vary between different LSU pairs or that LSUs associate forming multimeric complexes.

Additional structural modeling and spatial distribution of the electrostatic potential of LSU-LSU dimers reveal significant differences in homo- and heterodimer formation, suggesting that the dimer formation by LSU might have a regulatory function [12]. These analyses also suggested that dimers might bind to different molecular partners than monomeric forms [12]. Additionally, combined mutagenesis and Y2H analyses have identified that the conserved cysteine residues (C54) are not involved in the dimer stabilization and do not form S–S bridges between the monomers of LSUs. Because these cysteine residues are located on the surface of the protein and exposed to solvent, it has been proposed that they play a role in the interaction with the coiled-coil structure while they could also be involved in the recognition of protein interactors [12]. Consistently, LSU proteins were identified as protein hubs in high-throughput Y2H analyses that interrogated the Arabidopsis interactome [9]. Interestingly, despite their high similarity the LSU interactomes partially overlap [9,10], suggesting some degree of specialization. For example, from 100 protein interactors identified for LSU1 or LSU2, only 17 are shared by both LSU proteins [14].

LSU 1, 2, and 3 are able to interact with multiple partners, including pathogen effectors from *Pseudomonas syringae* and *Hyaloperonospora arabidopsidis* and proteins involved in different biological processes, including some related to plant immune processes [9]. From these interactors, the chloroplastic iron-dependent superoxide dismutase FSD2 has been independently validated as a partner for LSU1 and LSU2, and this interaction was shown to be relevant for stimulation of FSD2 activity and production of H_2_O_2_ [19]. Another study described MYB51, a transcription factor involved in glucosinolate biosynthesis as partner for LSU3 [36]. Further interactors for LSU1-4 were identified in vivo by using tandem affinity purification-mass spectrometry (TAP-MS) under sulfate sufficiency and deficiency conditions, and further confirmed by BiFC and Y2H [12]. Among confirmed proteins, ATP sulfurylase 1 (APS1), first enzyme in sulfate assimilation, the GRF8 transcription factor involved in plant growth, chlorophyll biosynthesis and seedling greening [37], the RAF2/SDIRIP1 protein involved in Rubisco assembly and ABA stress responses [38,39] and the GAPC1 C subunit of cytosolic GADPH enzyme (described as a redox switch with roles in plant responses to stress [40,41]) were validated by BiFC, whereas Y2H assays confirmed the interaction of LSUs and RAF2/SDIRIP1, the peroxisomal catalase 2 (CAT2) and the autophagy cargo receptor NBR1, involved in the crosstalk between autophagy and ABA signaling [42]. Interestingly, Y2H analysis in *Nicotiana plumbaginifolia* had identified a homolog of Arabidopsis NPR1, Joka2, as an interacting partner for UP9C [43], indicating a conserved function of LSUs and selective protein degradation by autophagy. Importantly, the strength of the interaction and protein partner varied among LSUs, and further Y2H analysis using LSUs with mutations in the conserved C54 cysteine residue further supported the idea that the shape of the dimer coiled coil is relevant for recognition of LSU targets [12].

These results are consistent with reported roles of LSUs in different aspects of abiotic and biotic stress responses in plants and with a specialization of LSU function in plants.

### 4.2. Crosstalk between LSUs and Phytohormones

Mapping of the Arabidopsis LSU interaction network [12] has revealed that the interacting partners of LSUs include diverse proteins, some of which have roles in hormonal signaling pathways. Y2H data support interactions of LSUs with JAZ1 and JAZ9, members of the JAZ (jasmonate ZIM-domain) family of repressors, the cytokinin B-type response regulator ARR14, and the ERF12 ethylene transcription factor [9,12]. In addition to these main signaling components of hormonal pathways, other proteins linked to hormonal responses have been described as LSU interactors, such as the GRF8 transcription factor, which is induced by brassinosteroids through BZR1 [44], the ZFP7 zinc finger, whose overexpression confers ABA insensitivity to seed germination [45], the H3K27 tri-methyltransferase SWN involved in the control of ABA-induced senescence-associated genes [46] or the EDS1 lipase, which promotes SA accumulation [47].

Furthermore, an interaction of UP9C and 1-aminocyclopropane-1-carboxylic acid (ACC) oxidase (ACO2A), the enzyme that catalyzes ethylene synthesis from ACC, has been reported in tobacco [13]. Short-period sulfur deficiency triggers an accumulation of ethylene, which is absent in antisense UP9C plants, indicating the interaction between UP9C and ACO2A is relevant for its function [13].

### 4.3. Differential Expression of Arabidopsis *LSU* Genes in Response to Phytohormone Treatment

Besides LSU interaction with proteins involved in hormonal pathways, evidence suggest that hormones can have a direct impact on LSU expression. The central regulator of the ethylene signaling pathway, EIN3, can bind the *LSU1* promoter *in vivo* and regulate its expression [48]. Specifically, a ChIP-qPCR assay showed that EIN3 protein bound strongly to fragments of *LSU1* promoter, and this result was confirmed by EMSA and yeast one-hybrid analyses [48]. Furthermore, a transient dual-luciferase assay in Arabidopsis protoplast indicated that EIN3 transcriptionally represses *LSU1*, which agrees with higher *LSU1* mRNA levels in *ein3-1* mutants [48]. These results demonstrate that phytohormone signaling pathways can regulate *LSUs* in Arabidopsis.

To get new insights into the response of *LSU* genes to phytohormones, we reviewed the transcriptomic data of Arabidopsis *LSUs* in Plant Regulomics database [49], which integrates 11,090 Arabidopsis transcriptomic datasets, including phytohormone treatments. As shown in Figure 3A, *LSU* genes significantly respond to at least one of the following phytohormones: ABA, ethylene, auxin, and jasmonate. *LSU1* is the member of this gene family with the highest number of experiments as a differentially expressed gene (adjusted *p*-value < 0.05) (eight experiments, Figure 3A). Specifically, *LSU1* is down-regulated by ethylene in three experiments, which is consistent with the previously reported repression of this gene by EIN3 [48]. In addition, *LSU1* is also downregulated by auxin (except in the experiment GSE1491) and jasmonate (Figure 3A). In contrast, *LSU2* and *LSU4* showed a positive response to auxin and the *LSU3* gene to jasmonate, suggesting that the response to phytohormones differs among members of the LSU family in Arabidopsis (Figure 3A).

These differences between members of the *LSU* family in response to phytohormones are also reflected in the regulatory network predicted for these genes (Figure 3B). The predicted TF-target interaction obtained from PlantRegMap [50] suggests that *LSU1* is mainly regulated by TFs from the ERF family, which are important regulatory components of ethylene signaling and are involved in plant development and stress responses by regulating the expression of ethylene-responsive genes [51]. On the contrary, *LSU2* is predicted to be regulated by ABA-associated TFs, such as ABF3, and *LSU3* by jasmonate-related TFs such as JAM2 (Figure 3B).

## 5. Conclusions

Although some information has been gathered about the *LSUs*, there are still many open questions about their functions. In this review, we have provided evidence that this group of proteins appears to display more multifaceted roles than previously expected.

Nevertheless, only a few plant LSU proteins have been functionally characterized. Members of the *LSU* family are likely to participate in fine-tuning responses to the different plant stresses, especially S limitation, and in various aspects of plant development, such as flowering and fruit formation. By modulating a variety of LSU in their protein–protein interactions, *LSU* might act in the crosstalk of various signaling pathways directly or indirectly linked to S metabolism.

It would be interesting to analyze the molecular mechanisms by which *LSUs* orchestrate metabolic homeostasis, plant stress responses, and plant growth and development. In this regard, we expect significant advances in connecting the structure and functions in this family of plant proteins in the following years.

The results accumulated in recent years in crops suggest that the *LSU* gene family could have significant implications for crop improvement in the future, particularly in regard to the S deficiency response and pathogen attack. The study of the *LSUs* in crops has the potential to lead to new strategies for crop improvement and sustainable agriculture.

## Figures and Tables

**Figure 1 ijms-24-02819-f001:**
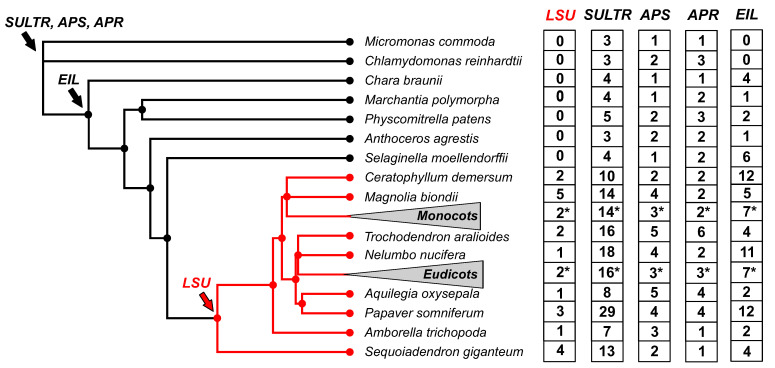
The *LSU* gene family appeared recently in plant evolution compared to other S-responsive genes. The phylogenetic tree was constructed according to [17]. To improve visualization, 73 eudicotyledon and 31 monocotyledon species collapsed in the phylogenetic tree (triangle), and the average number of *LSU* genes are indicated with an asterisk. The copy number of *LSUs*, sulfate transporters (*SULTR*), ATP sulfurylases (*APS*), *adenosine 5’-phosphosulfate reductase* (APR), and *ethylene-insensitive3-like* transcription factors (*EIL*) was obtained from the PLAZA 5.0 database [18].

**Figure 2 ijms-24-02819-f002:**
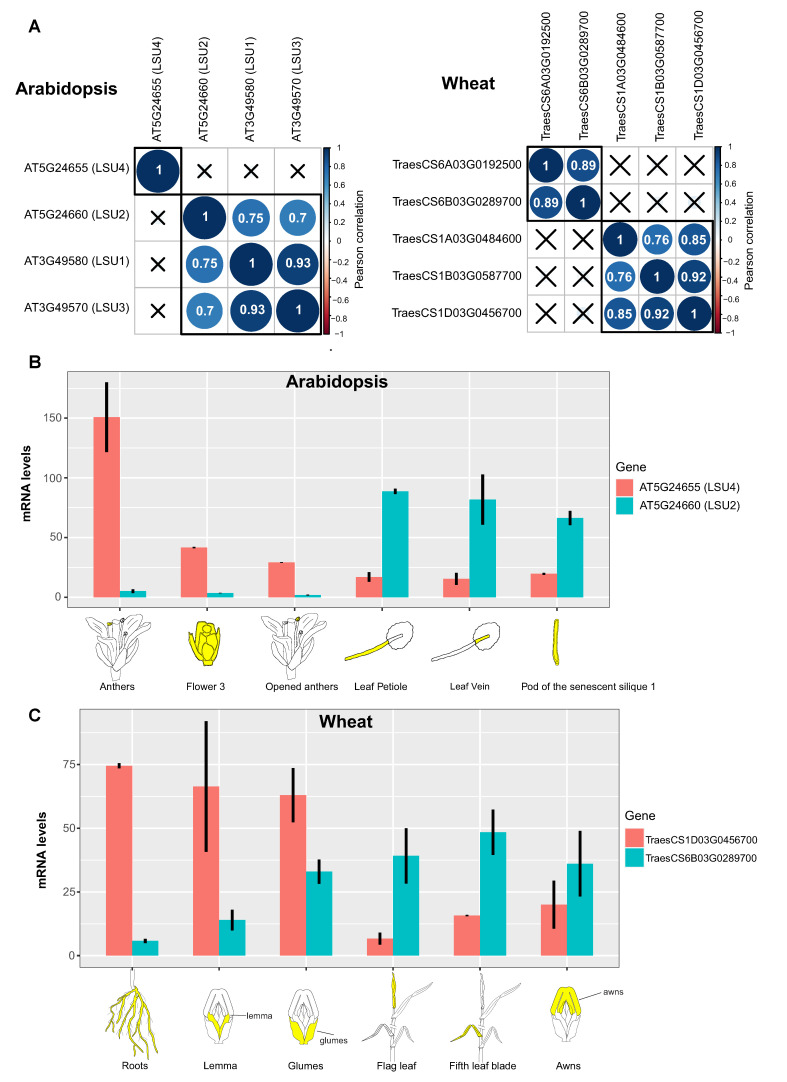
Analysis of the *LSU* developmental expression atlas in Arabidopsis and wheat suggests the existence of two LSU subgroups with contrasting expression profiles. (**A**) Coexpression analyses of Arabidopsis (left panel) and wheat LSU family (right panel) performed with the r package “*corrplot*” [23] using all samples included in the developmental atlas of ePlant [21]. Only Pearson correlation values with a *p*-value < 0.01 are shown in the correlation heatmaps. (**B**) Expression profiles of Arabidopsis *LSU4* and *LSU2* in the three samples with the higher expression of each selected gene. (**C**) Expression profiles of wheat *TraesCS1D03G0456700* and *TraesCS6B03G0289700* in the three samples with the higher expression of each selected gene.

**Figure 3 ijms-24-02819-f003:**
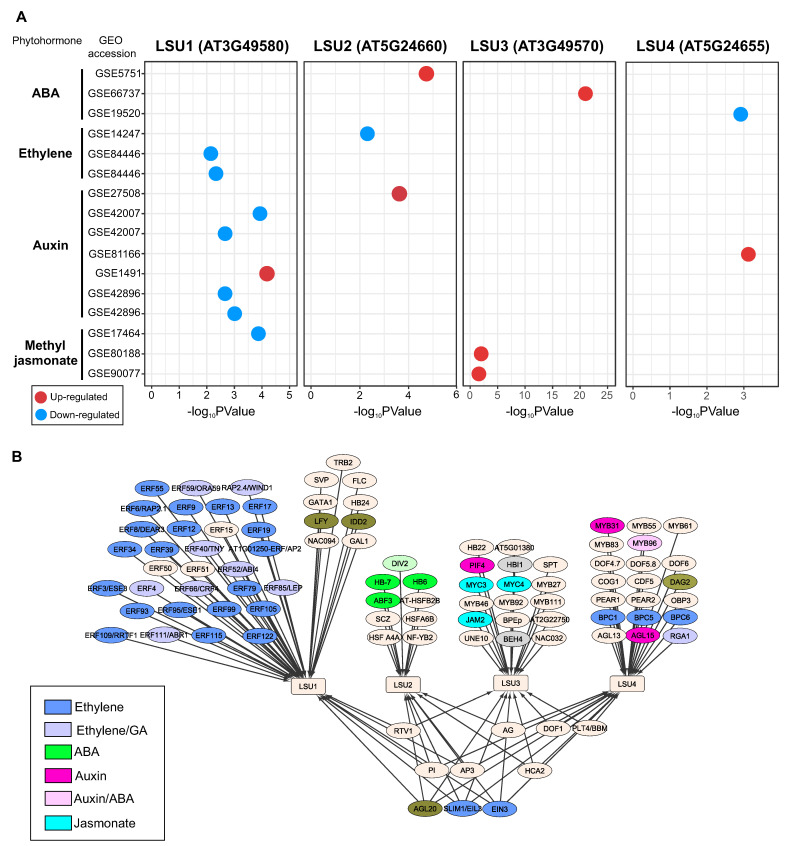
The Arabidopsis *LSU* genes significantly respond to different phytohormone treatments (**A**), and the analysis of gene regulatory network (**B**) predicted that contrasting groups of phytohormone-related transcription factors regulate each *LSU* gene. Identification of the significant response of *LSU* genes to phytohormone treatments was performed by using transcriptome datasets available in the Plant Regulomics database [49]. The regulatory interactions between *LSU* genes and transcription factors were obtained PlantRegMap [50].

**Table 1 ijms-24-02819-t001:** List of crop species with experimental evidence for *LSU* genes.

Crop Species	Gene ID	Biological Function	Tissue	Evidence	References
*Solanum lycopersicum*	Solyc03g096760.1	S-deficiency	Roots and leaves	RNA-seq, qPCR	[17,30]
	Solyc03g096770.1	S-deficiency	Roots and leaves	RNA-seq, qPCR	[17,30]
	Solyc03g096780.1	S-deficiency	Roots and leaves	RNA-seq, qPCR	[17,30]
	Solyc06g072990.1	S-deficiency	Roots and leaves	RNA-seq, qPCR	[17,30]
*Oryza sativa*	Os02g0129800	S-deficiency	Roots and shoots	RNA-seq	[31]
	Os10g0509401	S-deficiency	Roots and shoots	RNA-seq	[31]
	Os10g0509600	S-deficiency	Roots and shoots	RNA-seq	[31]
*Triticum aestivum*	TraesCS1D03G0456700	S-deficiency	Roots and leaves	qPCR	[17]
*Pisum sativum*	Psat5g070520	S-deficiency	Seeds	mRNA-protein network analysis	[32]
*Saccharum spontaneum*	S.offnewGene71819	Fungus infection	Stems	RNA-seq and qPCR	[33]
	Sspon.01G0026550-1A	Fungus infection	Stems	RNA-seq	[33]
	Sspon.01G0026550-2C	Fungus infection	Stems	RNA-seq	[33]
*Brassica juncea*	BjuA033112	Glucosinolate content	Seeds	GWAS	[34]

## Data Availability

Not applicable.

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
