# Peer review of "A Revised View of the LSU Gene Family: New Functions in Plant Stress Responses and Phytohormone Signaling"

_ijms, 2023, doi:10.3390/ijms24032819_

Round 1

Reviewer 1 Report

The manuscript “A revised view of the LSU gene family: New functions in plant stress responses and phytohormone signaling” by authors Canales et al. briefly summarize the current knowledge about the LSU gene family in plants and their potential roles in phytohormone responses. Overall, the most study about LSU was included in this review with some novel information regarding LSU gene family presented. However, most information of LSU from previous literates was just simply stated with limited further discussion. Especially some concerns should be paid attention as listed:

1. The evidence for LSU proteins involved in brassinosteroid signaling is very unreliable with the protein interaction of LSU and GRF8 (Line 42; Lines 270-272), although 14-3-3 proteins are known to be involved in cytoplasmic retention of BZR1/BES1. 14-3-3s are also important in multiple biological processes with a wide broad of targets. Also LSU in the study was found no response to BR other than all other phytohormones (Lines 295-299; Figure 3). Thus, role of LSU in BR signaling is still faint in current study.

2. In Line 73-74, the chromosomal distribution of LSU genes are wrong and not consistent with previous study (Sirko et al. The family of LSU-like proteins. Front Plant Sci. 2015 Jan 13;5:774. doi: 10.3389/fpls.2014.00774).

3. In line 183, please clarify the statement “no individual T-DNA lines exist for all LSU genes”; which is disagree with results Line 169, Line 174.

4. Line 67: delete ONE “LSU1/2/3”; References 7,10,12,14, 35 should be corrected.

Author Response

The manuscript “A revised view of the LSU gene family: New functions in plant stress responses and phytohormone signaling” by authors Canales et al. briefly summarize the current knowledge about the LSU gene family in plants and their potential roles in phytohormone responses. Overall, the most study about LSU was included in this review with some novel information regarding LSU gene family presented. However, most information of LSU from previous literates was just simply stated with limited further discussion. Especially some concerns should be paid attention as listed:

RESPONSE: Thank you for your feedback on our review of the LSU gene family. We apologize if you felt that the information from previous literature was not sufficiently discussed. In the new version of the manuscript, we have made some improvements in the discussion of several topics as phytohormones and protein interactions, and also included new section about LSU and crops. We are confident that these additions will provide a more comprehensive understanding of the LSU gene family and its role in plant growth and development.

  1. The evidence for LSU proteins involved in brassinosteroid signaling is very unreliable with the protein interaction of LSU and GRF8 (Line 42; Lines 270-272), although 14-3-3 proteins are known to be involved in cytoplasmic retention of BZR1/BES1. 14-3-3s are also important in multiple biological processes with a wide broad of targets. Also LSU in the study was found no response to BR other than all other phytohormones (Lines 295-299; Figure 3). Thus, role of LSU in BR signaling is still faint in current study.

RESPONSE: Thank you for the suggestion. It has been corrected in the new version of the manuscript.

  1. In Line 73-74, the chromosomal distribution of LSU genes are wrong and not consistent with previous study (Sirko et al. The family of LSU-like proteins. Front Plant Sci. 2015 Jan 13;5:774. doi: 10.3389/fpls.2014.00774).

RESPONSE: Thank you for noticing this erro. It has been corrected in the new version of the manuscript.

  1. In line 183, please clarify the statement “no individual T-DNA lines exist for all LSU genes”; which is disagree with results Line 169, Line 174.

RESPONSE: It is correct that no individual T-DNA lines exist for all LSU genes, because in the case of LSU1 no insertional mutant exists (https://doi.org/10.3389/fpls.2014.00774).

  1. Line 67: delete ONE “LSU1/2/3”; References 7,10,12,14, 35 should be corrected.

RESPONSE: Thank you for noticing this error. It has been corrected in the new version of the manuscript.

Reviewer 2 Report

The manuscript ‘A revised view of the LSU gene family: New functions in plant stress
responses and phytohormone signaling’ by Canales et al. describe the overview of the functional response of the low Sulphur (LSU) gene family to phytohormone signaling and stress response in plants. The manuscript is well-conceptualized and well-written. I have come across a few reviews/articles on the LSU gene family published recently, which cover the functional mechanisms in various species. I have a few observations on this manuscript which may be useful for further improvement;

The author may enlist the crop-wise genes of the LSU family, their primers (if available), and functional mechanisms along with the citation in a tabular form to elucidate the updated work done so far. Sulphur is an important component of oilseed crops. The role of the LSU gene family may be highlighted in response to the oilseed crops.

The functional role of the LSU gene family in biotic and abiotic stresses, along with hormonal signaling, needs to be specified crop-wise.

Line 20: Citations [1]; [3] may be written as [1,3]. Please revise throughout the text.

Line 48: ‘The molecular functions of LSU proteins are currently unknown’…I think the function of the LSU protein gene is well-documented..Please revise the sentence as limited, or scanty.

Any information on functional validation of the LSU genes for biotic and abiotic stress tolerance may be highlighted.

Line 153: Please revise lsu2 as LSU2…Please revise throughout the text.

Line 157: P.syringae ….give a space in between

Line 195: ‘LSU1 and LSU2’…Please mention all the genes in italics throughout the text.

Please elaborate on the future thrust/implications of LSU gene family in crop improvement in the Conclusion section.

References may be arranged in the proper format.

Language, punctuation, and grammar may be cross checked.

Overall, this is a well-written manuscript that contains valuable information and may be considered for publication with minor revision.

Good luck with the revision.

Author Response

The manuscript ‘A revised view of the LSU gene family: New functions in plant stress
responses and phytohormone signaling’ by Canales et al. describe the overview of the functional response of the low Sulphur (LSU) gene family to phytohormone signaling and stress response in plants. The manuscript is well-conceptualized and well-written. I have come across a few reviews/articles on the LSU gene family published recently, which cover the functional mechanisms in various species. I have a few observations on this manuscript which may be useful for further improvement;

The author may enlist the crop-wise genes of the LSU family, their primers (if available), and functional mechanisms along with the citation in a tabular form to elucidate the updated work done so far. Sulphur is an important component of oilseed crops. The role of the LSU gene family may be highlighted in response to the oilseed crops.

RESPONSE: Thank you for the suggestion. In the new version of the manuscript, we have incorporated the information about LSU in crops in the section 3.4 and in the new table1.  

The functional role of the LSU gene family in biotic and abiotic stresses, along with hormonal signaling, needs to be specified crop-wise.

RESPONSE: This is a good point. We have included a new section about LSU gene family in crops (section 3.4)

Line 20: Citations [1]; [3] may be written as [1,3]. Please revise throughout the text.

RESPONSE: Thank you for noticing this format error, we have corrected in the new version of the manuscript.

Line 48: ‘The molecular functions of LSU proteins are currently unknown’…I think the function of the LSU protein gene is well-documented. Please revise the sentence as limited, or scanty.

RESPONSE: We have corrected in the new version of the manuscript by this sentence: “The molecular functions of LSU proteins remain incompletely understood”.

Any information on functional validation of the LSU genes for biotic and abiotic stress tolerance may be highlighted.

RESPONSE: Thank you for the suggesting, it has been included in the section 3.4.

Line 153: Please revise lsu2 as LSU2…Please revise throughout the text.

RESPONSE: In this case does not correspond italics because is referred to the protein.

Line 157: P.syringae ….give a space in between

RESPONSE: Corrected.

Line 195: ‘LSU1 and LSU2’…Please mention all the genes in italics throughout the text.

RESPONSE: In this case does not correspond italics because is referred to the protein.

Please elaborate on the future thrust/implications of LSU gene family in crop improvement in the Conclusion section.

RESPONSE: Thank you for the suggestion. In the new version of the manuscript, we have incorporated the implications of LSUs in crops.

References may be arranged in the proper format.

RESPONSE: Thank you for noticing this format error, we have corrected in the new version of the manuscript.

Language, punctuation, and grammar may be cross checked.

RESPONSE: Thank you for your suggestion. We make sure to cross-check the language, punctuation, and grammar in the new version of the manuscript.
